# Synergistic potential of lopinavir and azole combinational therapy against clinically important *Aspergillus* species

**Nicolas Burns**[1,2], **Ehab A. Salama**[1,2], **Mohamed N. Seleem**[1,2]*

**1** Department of Biomedical Sciences and Pathobiology, Virginia-Maryland College of Veterinary Medicine, Virginia Polytechnic Institute and State University, Blacksburg, Virginia, United States of America, **2** Center for One Health Research, Virginia Polytechnic Institute and State University, Blacksburg, Virginia, United States of America

* seleem@vt.edu

**Data Availability Statement:** All relevant data are within the manuscript. There are no other data sets within the manuscript that are not present within the manuscript. Data has been saved to harvard dataverse: https://doi.org/10.7910/DVN/JEVIOV.

## Abstract

*Aspergillus fumigatus* is a widely distributed pathogen responsible for severe infections, particularly in immunocompromised individuals. Triazoles are the primary treatments options for *Aspergillus* infections; however, the emergence of acquired resistance to this antifungal class is becoming a growing concern. In this study, we investigated the potential of the antiviral drug, lopinavir (LPV) to restore the susceptibility of *A. fumigatus* strains to a set of azoles, while also reducing the required azole dosage for treatment of susceptible isolates. The combination of LPV with either itraconazole (ITC) or posaconazole (POS) demonstrated potent synergistic interactions against 16 out of 23 (~70%) and 21 out of 23 (~91%) *A. fumigatus* isolates, respectively. Moreover, the combination showed synergistic activity against other clinically important *Aspergillus* species, including *A. niger*, *A. flavus*, and *A. brasiliensis*. The fractional inhibitory concentration index (FICI) for the combinations ranged from 0.18 to 0.313 for ITC and 0.091 to 0.313 for POS, indicating strong synergistic effects. Further investigation revealed that efflux pump inhibition contributed to the synergy observed between azole and LPV. Morphological examination of the fungal cells subjected to this combinational therapy at sub-inhibitory doses showed the presence of carbohydrate granules/patches. The identification of LPV as a promising adjunct therapy holds promise for addressing the emerging challenge of azole resistance in *Aspergillus* species and improving treatment outcomes for patients.

## Introduction

*Aspergillus fumigatus* is a filamentous, spore-forming fungus posing a significant threat to immunocompromised individuals across variety of settings [1, 2]. The fungus itself is ubiquitous in nature and human society [3]. Widely distributed in nature and human environments, *A. fumigatus* can cause various infections, including allergic bronchopulmonary aspergillosis, cutaneous aspergillosis, chronic pulmonary aspergillosis, aspergilloma (fungal ball), and invasive aspergillosis. Chronic pulmonary aspergillosis alone affects an estimated 3 million people

**Funding:** The author(s) received no specific funding for this work.

**Competing interests:** The authors have declared that no competing interests exist.

globally [4]. With an increasing number of immunocompromised patients, the incidence of *A. fumigatus* infections has risen significantly, resulting in over 200,000 infections annually, with mortality rates reaching as high as 90% [5, 6].

Addressing the urgent need for effective treatments, there is a demand for drugs capable of restoring the activity of azoles against *A. fumigatus*. Combinational therapies, like voriconazole and anidulafungin, have shown success in treating severe fungal infections [7, 8]. However, novel antifungal development is hampered due to the structural similarities between fungal drug targets and human proteins, leading to limited viable options and potential off-target effects. Although antifungals such as amphotericin B are effective, patients often endure severe side effects and may discontinue treatment.

*Aspergillus* species frequently encounter multiple azole fungicides in agriculture and clinical settings, contributing to the development of azole resistance [9, 10]. In *A. fumigatus*, azole resistance commonly occurs with a combination of alterations in the cyp51A gene, including tandem repeats (TR34 or TR46) and nonsynonymous point mutations (L98H/Y121F/T289A) [8, 11, 12]. Additionally, efflux pump machinery plays a significant role in triazole resistance, and investigations have revealed its concerning ability to readily adapt [13–15]. Non-Cyp51A triazole resistance, driven by efflux pump overactivity, is a growing concern, as direct overexpression has been shown to increase drug resistance [15, 16]. These mounting challenges, combined with limited novel antifungals options, have become a paramount concern for health organizations such as the Centers for Disease Control and Prevention (CDC) and the World Health Organization (WHO) [17].

In this study, our main objective was to identify new drugs that can be repurposed from their originally approved therapy to treat *Aspergillus* species infections. Building on our previous work, screening libraries for compounds/drugs that enhance the activity of azole drugs against Candida [18–21], we focused on LPV and antiviral agent as an enhancer of azoles against *Aspergillus* species. Lopinavir, a protease inhibitor originally developed as an antiretroviral drug for treating HIV/AIDS, has garnered significant attention as a potential repurposed therapeutic in various other medical contexts. Its FDA approval since highlights its established profile, enhancing its potential as a co-drug candidate for managing invasive fungal infections. Additionally, we explored the impact of the LPV/azole combination on the hyphal formation, a critical step for fungal dissemination. To gain insights into the mechanism of action, we investigated the ability of LPV, known to be poor ABC substrate, to interfere with the efflux pump machinery of *A. fumigatus* [22]. Our work seeks to highlight LPV's potential role in efflux inhibition, a crucial mechanism for antifungal resistance, facilitating the synergy observed between POS/ITC and LPV. The combination of potent *in vitro* synergy with azoles and the ability to interfere with fungal efflux pump makes LPV an exceptionally promising drug of high interest in this study. These findings represent a significant step towards identifying alternative treatments for *Aspergillus* species infections, and the repurposing of LPV as a potential adjunct therapy with azoles offers a new avenue to combat the challenge of antifungal resistance in these pathogens.

## Results

### Minimum inhibitory concentration (MIC) and microdilution checkerboard

To determine the MICs for ITC, POS, VRC and LPV against 27 different *Aspergillus* isolates, we perfromed three independent assays following CLSI protocol M38 [23]. The antifungal activity, complete growth inhibition, of itraconazole ranged from 0.25 to 64 μg/mL, while posaconazole and voriconazole ranged from 0.25 to 2 μg/mL [24]. CLSI reference isolates, MYA-3626 and -3627 MIC values for ITC, POS, and VRC were determined to be within

**Table 1. Description of the fungal strains utilized in this study and their minimum inhibitory concentration (MICs).**

| *Aspergillus* Strain/Designation | MIC [μg/mL] ITC | MIC [μg/mL] POS | MIC [μg/mL] VRC | MIC [μg/mL] LPV | Source & Characteristic & Resistance Mechanism |
|---|---|---|---|---|---|
| NR-35301 | 1 | 0.5 | 0.125 | >256 | Human abdominal tissue (1998) |
| NR-35302 | 1 | 0.5 | 0.25 | >256 | Human peritoneal fluid (1998) |
| NR-35303 | 1 | 0.5 | 0.25 | >256 | Human sputum-tracheal suction(1998) |
| NR-35304 | 1 | 0.5 | 0.25 | >256 | Human sputum-tracheal suction(1998) |
| NR-35305 | 1 | 0.5 | 0.25 | >256 | Hospital Environment (1998) |
| NR-35307 | 1 | 0.5 | 0.25 | >256 | Hospital Environment (1998) |
| NR-35310 | 1 | 0.5 | 0.25 | >256 | Environmental Isolate (2002) |
| NR-35311 | 1 | 0.5 | 0.25 | >256 | Environmental Isolate (2002) |
| NR-35312 | 1 | 0.5 | 0.25 | >256 | Environmental Isolate (2002) |
| NR-41311 | 1 | 0.5 | 0.5 | >256 | Human Sputum |
| NR-41312 | 1 | 0.5 | 0.125 | >256 | Human Sputum |
| CDC 731 | 64 | 1 | 2 | >256 | L98H, TR34 |
| CDC 732 | 64 | 1 | 1 | >256 | F495I, L98H, S297T, TR34 |
| CDC 733 | 64 | 1 | 2 | >256 | L98H, TR34 |
| CDC 734 | 64 | 1 | 2 | >256 | L98H, TR34 |
| CDC 735 | 64 | 1 | 1 | >256 | F495I, L98H, S297T, TR34 |
| CDC 736 | 1 | 1 | 0.25 | >256 | USA |
| CDC 737 | 1 | 1 | 0.25 | >256 | USA |
| CDC 738 | 1 | 1 | 0.5 | >256 | USA |
| CDC 739 | 1 | 1 | 0.5 | >256 | USA |
| CDC 740 | 1 | 1 | 0.5 | >256 | USA |
| MYA-3626 | 2 | 2 | 1 | >256 | USA |
| MYA-3627 | >16 | 2 | 1 | >256 | USA |
| *Aspergillus niger* 6275 | 2 | 2 | ND | >256 | Isolate from leather |
| *Aspergillus niger* 16888 | 2 | 1 | ND | >256 | |
| *Aspergillus flavus* 9643 | 0.5 | 2 | ND | >256 | Isolate from New Guinea |
| *Aspergillus brasiliensis* 16404 | 8 | 2 | ND | >256 | Isolate from blueberry, NC, USA |

\* Minimum Inhibitory Concentration (MIC), in which no observable growth was identified.

\*\* Lopinavir (LPV), Itraconazole (ITC), Posaconazole (POS), Voriconazole (VRC), and Not Determined (N.D.)

established ranges [23, 25]. Notably, five isolates displayed resistance to azoles, with MICs of 64, 1, and ~1 μg/mL for ITC, POS and VRC, respectively. LPV, on the other hand, did not exhibit antifungal activity up to 256 μg/mL (Table 1).

To identify synergistic interactions between azole antifungals (ITC, POS, and VRC) and LPV, a checkerboard assay was performed. We observed synergy between ITC and LPV in 16/21 strains with ΣFICI values ranging from 0.137–0.313 (Table 2). Similarly, for POS, we observed synergy in 19/21 strains with ΣFICI values ranging from 0.091–0313 (Table 2). However, the interaction between VRC and LPV was indifferent against all tested strains, with ΣFICI values ranging from 0.563–1.125 (Table 2). Surprisingly, we observed synergy in three POS-resistant strains, CDC #731 #733 and #734 with ΣFICI values of 0.281, 0.312, and 0.312 (Table 2). Furthermore, were examined *A. brasiliensis*, *A. flavus* and *A. niger* via checkerboard assay to evaluate the efficacy of LPV in enhancing antifungal activity of azoles against other *Aspergillus* species. The results demonstrated synergy for all the strains treated with POS/ITC in combination with LPV, ΣFICI values of 0.265–0.281 and 0.091–0.312, respectively

**Table 2. Synergistic activity of lopinavir (LPV) and azole antifungals against different strains of *Aspergillus*.**

| *A. fumigatus* | LPV/ITC Combination | | | LPV/POS Combination | | | LPV/VRC Combination | | |
|---|---|---|---|---|---|---|---|---|---|
| | MIC [μg/mL] | ΣFICI | Mode | MIC [μg/mL] | ΣFICI | Mode | MIC [μg/mL] | ΣFICI | Mode |
| NR-35301 | 16/0.125 | 0.156 | SYN | 16/0.06 | 0.151 | SYN | 0.25/0.25 | 1.001 | IND |
| NR-35302 | 16/0.125 | 0.156 | SYN | 32/0.06 | 0.183 | SYN | 0.25/0.5 | 1.001 | IND |
| NR-35303 | 8/0.125 | 0.141 | SYN | 16/0.03 | 0.091 | SYN | 32/0.25 | 0.625 | IND |
| NR-35304 | 8/0.125 | 0.141 | SYN | 32/0.06 | 0.183 | SYN | 0.25/0.5 | 1.001 | IND |
| NR-35305 | 16/0.125 | 0.156 | SYN | 8/0.125 | 0.266 | SYN | 32/0.25 | 1.125 | IND |
| NR-35307 | 32/0.125 | 0.188 | SYN | 8/0.125 | 0.266 | SYN | 0.25/0.5 | 1.001 | IND |
| NR-35310 | 32/0.125 | 0.188 | SYN | 8/0.125 | 0.266 | SYN | 32/0.25 | 1.125 | IND |
| NR-35311 | 16/0.125 | 0.156 | SYN | 16/0.125 | 0.281 | SYN | 32/0.25 | 1.125 | IND |
| NR-35312 | 32/0.125 | 0.188 | SYN | 16/0.125 | 0.281 | SYN | 16/0.5 | 1.063 | IND |
| NR-41311 | 16/0.125 | 0.156 | SYN | 16/0.125 | 0.281 | SYN | 0.25/0.25 | 1.001 | IND |
| NR-41312 | 16/0.125 | 0.156 | SYN | 16/0.125 | 0.281 | SYN | 32/0.25 | 1.125 | IND |
| CDC 731 | 32/16 | 0.313 | SYN | 16/0.5 | 0.281 | SYN | 0.25/4 | 1.001 | IND |
| CDC 732 | 32/64 | 1.063 | IND | 32/1 | 0.563 | IND | 0.25/1 | 1.001 | IND |
| CDC 733 | 32/64 | 1.063 | IND | 32/0.5 | 0.313 | SYN | 0.5/4 | 1.002 | IND |
| CDC 734 | 32/64 | 1.063 | IND | 32/0.5 | 0.313 | SYN | 0.25/4 | 1.001 | IND |
| CDC 735 | 32/64 | 1.063 | IND | 32/1 | 0.563 | IND | 0.25/1 | 1.001 | IND |
| CDC 736 | 8/0.5 | 0.516 | IND | 16/0.25 | 0.281 | SYN | 16/0.25 | 0.563 | IND |
| CDC 737 | 16/0.25 | 0.281 | SYN | 16/0.125 | 0.281 | SYN | 0.25/0.25 | 1.001 | IND |
| CDC 738 | 16/0.25 | 0.281 | SYN | 16/0.25 | 0.281 | SYN | 16/0.25 | 0.563 | IND |
| CDC 739 | 16/0.25 | 0.281 | SYN | 16/0.125 | 0.156 | SYN | 32/0.25 | 0.625 | IND |
| CDC 740 | 8/0.25 | 0.266 | SYN | 16/0.125 | 0.156 | SYN | 16/0.25 | 0.563 | IND |
| MYA-3626 | 32/2 | 1.063 | IND | 16/0.5 | 0.281 | SYN | 32/0.5 | 0.563 | IND |
| MYA-3627 | 32/16 | 1.063 | IND | 16/0.5 | 0.281 | SYN | 32/0.5 | 0.563 | IND |
| *Aspergillus niger* 6275 | 8/0.5 | 0.266 | SYN | 32/0.125 | 0.125 | SYN | NA | NA | NA |
| *Aspergillus niger* 16888 | 8/0.5 | 0.266 | SYN | 16/0.06 | 0.091 | SYN | NA | NA | NA |
| *Aspergillus flavus* 9643 | 16/0.125 | 0.281 | SYN | 16/0.25 | 0.156 | SYN | NA | NA | NA |
| *Aspergillus brasiliensis* 16404 | 8/2 | 0.266 | SYN | 16/0.5 | 0.281 | SYN | NA | NA | NA |

***FICI (Fractional Inhibitory Concentration Index) utilized to quantify interactions between the tested combinations with the following definitions: Synergy (SYN) is an ΣFICI values ≤ 0.5, Indifference (IND) is an ΣFICI of > 0.5 to ≤ 4, and Antagonistic with an ΣFICI value of > 4.

**** Lopinavir (LPV), Itraconazole (ITC), Posaconazole (POS), Voriconazole (VRC), and Not Applicable (N/A)

(Table 2). Additionally, we investigated the possibility of reducing the necessary azole concentration further by applying a fixed concentration of LPV (16 μg/mL) in the media (Fig 1A–1C). When 16 μg/mL of LPV was utilized in the media, 50% of the CDC isolates screened with ITC and LPV demonstrated reduced MICs (Fig 1A and 1B). VRC displayed no significant reductions (Fig 1C). Moreover, all strains screened with POS and LPV showed a reversion from resistant profiles (Fig 1A and 1B).

## Efficacy of LPV and azole combinational treatment on the growth kinetics of *A. fumigatus* clinical isolate CDC #738

To investigate the effect of azole/LPV combination on the growth kinetics of *A. fumigatus*, we conducted a growth kinetics assay using *A. fumigatus* strain CDC #738. The fungal strain, 2.5 x10$^5$, was treated with LPV (16 μg/mL), ITC (0.25 μg/mL), VRC (0.125 μg/mL) and POS (0.125 μg/mL), and their combinations. As observed, LPV, POS, VRC and ITC alone did not have a significant impact on fungal growth and closely resembled the untreated control

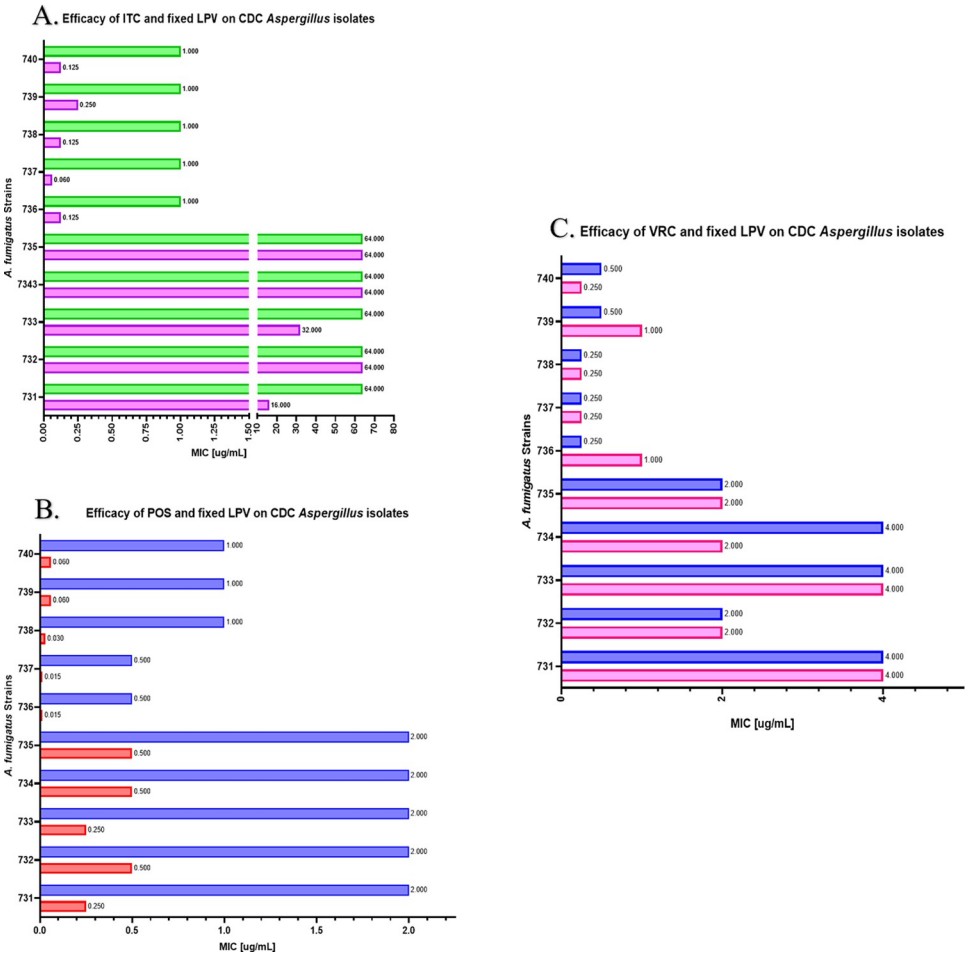

**Fig 1. Efficacy of fixed LPV combinational treatment on the susceptibility of *A*. *fumigatus* clinical isolates to itraconazole(A)/posaconazole(B)/voriconazole(C).** Lopinavir at a fixed concentration of 16 μg/mL with serially diluted posaconazole and itraconazole displays potent reduction in MIC when present. Red/Purple/Magenta contain LPV, while Blue/Green/Blue lack LPV.

(Fig 2). Fungal growth was not inhibited through the co-application of LPV and VRC. However, when treated with ITC and LPV, fungal growth was arrested for nearly 30 hr until the azole become tolerated. Similarly, the combination of POS and LPV resulted in arrested fungal growth for 48 hr.

## Effect of LPV on efflux machinery

Efflux pumps play a critical role in microorganisms by facilitating the removal of undesirable products, including antifungal drugs, thereby contributing to drug resistance [14]. To investigate the mechanism by which LPV enhances azole activity, we evaluated the ability of LPV to inhibit efflux in *Aspergillus* isolates. Using a Rhodamine-6G efflux assay with the addition of glucose, we observed significant differences between untreated and treated groups. As observed, LPV is significantly capable of hindering *A*. *fumigatus* efflux machinery up to 70% when compared to the untreated cells ($P<0.001$)(Fig 3). These results demonstrate that LPV can significantly disrupt the function of essential efflux mechanisms that fungi use to expel toxic dyes (e.g. R6G) and antifungal agents (e.g. azoles).

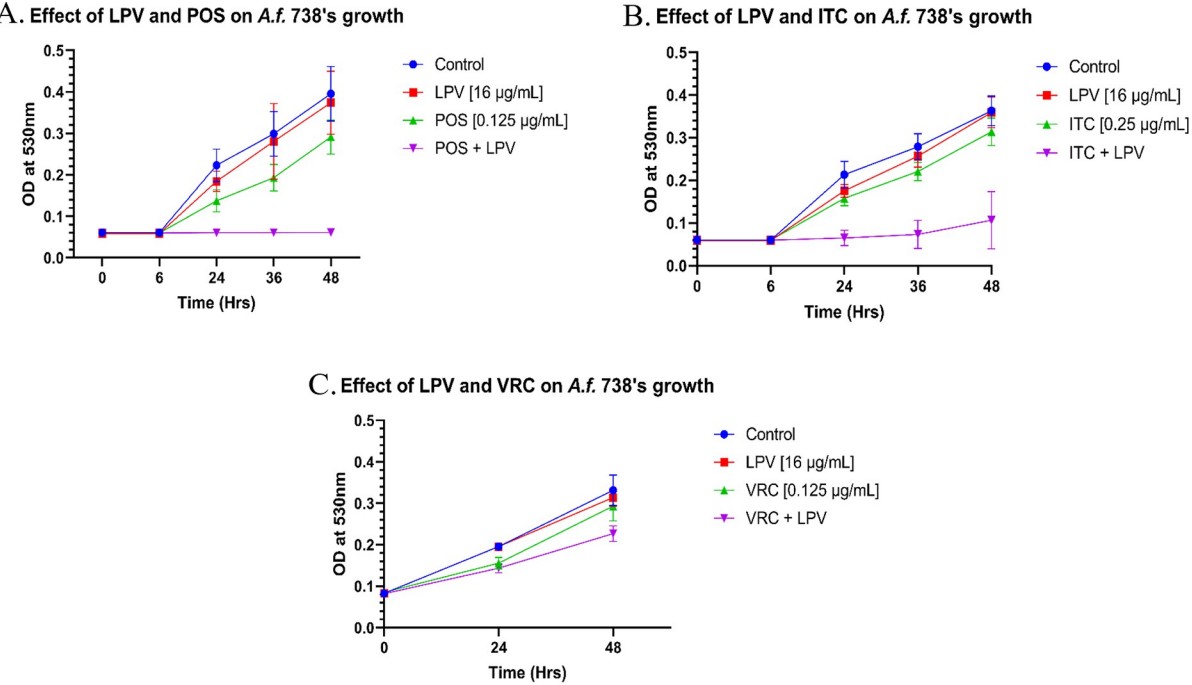

**Fig 2. Effect of POS/LPV and ITC/LPV on the growth kinetics of *A. fumigatus* strain CDC #738. (A)** Growth kinetics of *A. fumigatus* CDC #738 in RPMI 1640 at 37˚C with or without LPV, POS, or both present in the media, OD at 530 nm was recorded over 48 hr. **(B)** Growth kinetics of isolate CDC #738 in RPMI 1640 treated with LPV, ITC or both present in the media. **(C)** Growth kinetics of isolate CDC #738 in RPMI 1640 treated with LPV, VRC or both present in the media. Data represents the means and standard deviations of triplicate determinations.

## Impact of LPV on formation of cell-wall carbohydrate patches induced by azoles

We aimed to investigate the impact of LPV on the formation of cell-wall carbohydrate patches induced by azoles When fungal cells are challenged with effective azole levels, they undergo cellular dysregulation, resulting in the formation of patches that can be easily stained by calco-fluor-white [26]. Untreated cells showed no aberrant morphology, including the absence of carbohydrate patches (Fig 4A). Cells challenged with LPV alone displayed similar properties to the untreated group (Fig 4B). Similarly, cells challenged with sub-inhibitory doses of either azole alone also showed minimum to no patch formation (Fig 4C and 4D). However, when challenged with ITC and LPV, patches were readily observed along the interior of hyphae (Fig 4E). This observation was consistent with cells treated with POS and LPV (Fig 4F). Interestingly, a decrease in septa was observed around the patches. Demonstrating ineffective azoles doses when applied alone results in negligible patch formation while through the co-application of LPV large groups can be observed.

## Discussion

Clinical outcomes of azole-resistant Aspergillosis are frequently unfavorable, resulting in treatment failures and elevated mortality rates [27]. In recognition of this significant concern, the WHO has designated *Aspergillus sp*. as a critical priority pathogen [27]. The scarcity of new antifungal options further accentuates the need for innovative treatment strategies. In response to these challenges, our study has delved into alternative therapeutic avenues, with a specific focus on drug repurposing. The concept of repurposing drugs has gained prominence in

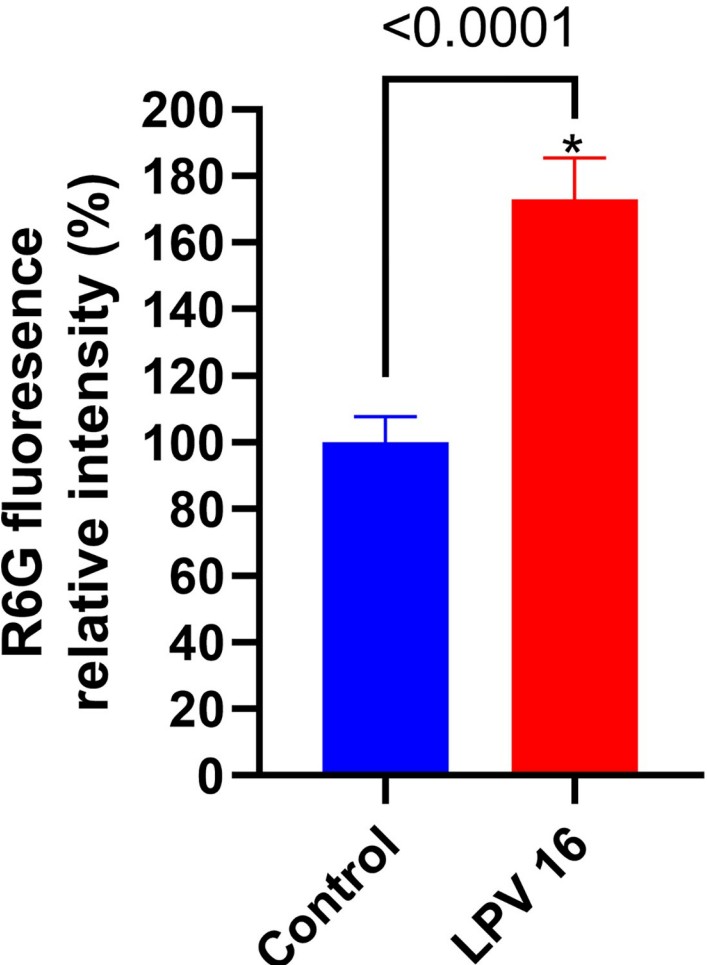

**Fig 3. Efflux pump inhibition in *A. fumigatus* by lopinavir.** R6G relative fluorescence in the presence or absence of LPV using *A. fumigatus* strain CDC #738. The relative change in the effluxed rhodamine as compared to the untreated, positive, control was determined as percentage. Data represents the means and standard deviations of triplicate determinations. * Indicates a significant difference between LPV and the untreated control (P < 0.0001) as determined by unpaired two-tailed *t*-Test.

antifungal drug discovery, proving particularly effective in identifying co-drugs that complement existing antimicrobial agents [18–20, 28, 29]. In a whole-cell screening assay of approximately 1500 FDA approved drugs we previously identified LPV, HIV protease inhibitor, as a promising candidate against *Candida* [20]. The rationale behind repurposing LPV as an antifungal agent lies in its capacity to effectively inhibit the efflux pump in *Candida*, which addresses a significant concern in the context of azole-resistant aspergillosis [18]. As shown by enhanced drug retention within the pathogen, we can effectively increase the availability of inhibitory concentrations of azole drugs at the target site. However, when considering modified drug targets, particularly for CDC isolates #731–735, a mixed effect is observed, as indicated in Tables 1 and 2. Our findings show that in the case of ITC isolate #731, synergy can be achieved with cyp51A mutations, specifically TR34 and L98H. Upon investigating the synergy of A. fumigatus #731, 733, and 734 with POS in combination with LPV, we observe that only TR34 and L98H mutants exhibit synergy, with one reverting to a susceptible profile. However, as we delve deeper into the analysis, we find that the emergence of additional mutations, such as F495I and S297T, leads to a failure in combination efficacy. This discovery prompted a

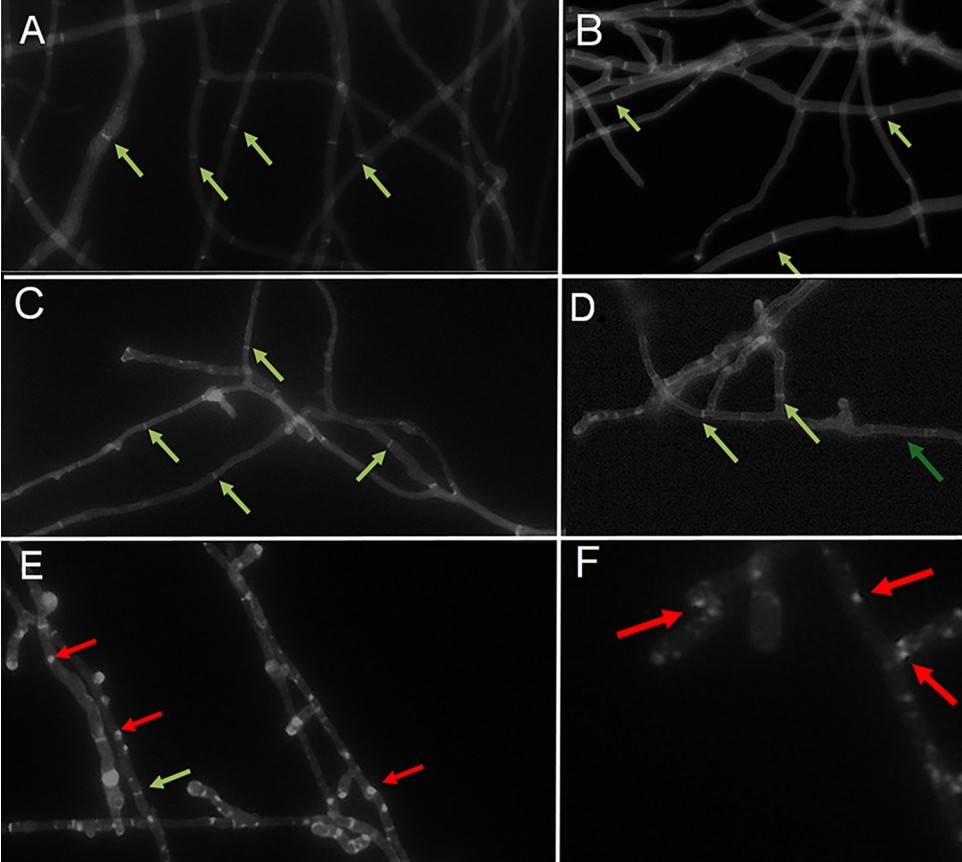

**Fig 4. Determination of toxic patch formation within hyphae of *A*. *fumigatus* arising from combinational therapy.** Figures A-F represent *A*. *fumigatus* grown (20hr) under different conditions. RPMI alone **(A)**, LPV [16 μg/mL] **(B)**, POS [0.125 μg/mL] **(C)**, ITC [0.25 μg/mL] **(D)**, LPV [16μg/mL] + POS [0.125 μg/mL] **(E),** and LPV [16 μg/mL] + ITC [0.25 μg/mL] **(F)**. The presence of septa, identified by green arrows, can be observed as bars spanning across the sides of the fungi. **(A)** & **(B)** show no other morphological oddities within the fungal cells, indicative of healthy unimpeded growth. These are also observed within **(C)** & **(D)** when sub-inhibitory of azole doses are applied to the fungus. Figures **(E)** & **(F)** represent *A*. *fumigatus* grown with a combination of LPV+POS **(E)**, LPV+ITC **(F)**; in these figures, green arrows can be observed to point out septa, a common motif. The red arrows point out morphological oddities not present in normal fungal cells. These granules or patches are conglomerations of carbohydrates resulting from dysregulation within the cell wall synthesis process.

comprehensive investigation into the potential synergistic effects of LPV in combination with three azoles against a diverse array of isolates from both resistant and susceptible *Aspergillus* species. Checkerboard data clearly illustrates the ability of LPV to synergize with ITC and POS, against various *Aspergillus* isolates. However, unlike its efficacy against *Candida*, the combination of LPV with VRC does not demonstrate the same degree of synergistic activity against *Aspergillus* isolates [18]. This disparity could be attributed to the unique interaction between the side chains of ITC and POS, which occupy a specific channel within CYP51 [29, 30]. This interaction, absent with VRC, is believed to bolster the stability of binding to CYP51 proteins, potentially explaining the differential response observed between the three azoles [29, 30].

Our initial checkerboard data demonstrated that the co-application of LPV enhances azole activity against susceptible strains, as well as two CDC-resistant strains Surprisingly, LPV was able to revert azole resistance in two resistant strains, CDC #731 and CDC #733, specifically

for POS. These findings underscore the promising role of LPV as a synergistic agent in combating azole resistance in *Aspergillus sp*.

Growth kinetic assays are crucial for understanding the ability of the microorganism to survive and proliferate in their environment. We demonstrate that both the azoles, ITC, VRC and POS, and LPV alone fail to inhibit fungal proliferation at sub-MIC concentrations. When POS and LPV are combined, we observe a significant reduction in fungal proliferation at sub-MIC concentrations over a 48-hour period. ITC and LPV, on the other hand, exhibited substantial inhibition of fungal growth up to 36 hours (Fig 2). In contrast, VRC and LPV failed to inhibit fungal growth at the same time points when ITC/POS and LPV were able to effectively curb growth. This underscores the immense promise of synergistic combinations as innovative therapeutic strategies in combating multidrug-resistant fungal infections.

The checkerboard data and growth kinetic assays have provided compelling insights into the enhanced activity of azoles against *Aspergillus sp*., igniting the exploration of underlying mechanisms. Among these mechanisms, efflux systems have emerged as pivotal players in triazole resistance within *Aspergillus* species [16]. These efflux mechanisms operate through the active expulsion of azoles from fungal cells, a process that diminishes intracellular drug concentrations and diminishes azole efficacy. Efflux pump genes, such as Cdr1B efflux transporter, ABC transporters (AtrF, AtrI), and a major facilitator superfamily transporter (MdrA) have been documented to exhibit elevated activity in azole-resistant clinical isolates of *Aspergillus sp*. [31]. Furthermore, cellular efflux assays have shown that LPV is unable to be effluxed via efflux pumps, including ABC transporters, within MCF-7/Dox40 and MCF-7/MR cells lines [22]. Additionally, LPV has been found to inhibit efflux activity of *C. auris* [20]. Given the similarity in functionality of ABC transporters to *Candida species*, which have been shown to contribute to resistance, we hypothesized that LPV application reduces the ability of *Aspergillus* to efflux azoles. The cells treated with LPV retained 70% more of Rhodamine-6G than the untreated group. This impressive retention rate has been observed when clorgyline, a proposed efflux pump inhibitor, was applied by Esquivel et al., [32]. Our results and those of Esquivel et al., collectively indicate that the *A. fumigatus* efflux system can be effectively inhibited by the addition of LPV or another efflux inhibitor, even with the addition of glucose [32]. While ITC and POS can be enhanced by efflux inhibition VRC demonstrated little increased efficacy. Literature suggests that VRC is not significantly impacted by efflux mechanisms, although it acts as a substrate and its presence can increase efflux activity [32, 33]. The intricate interplay between LPV and efflux mechanisms holds promise for the development of novel strategies to counteract azole resistance in *Aspergillus* isolates, thereby advancing our understanding of potential avenues for antifungal intervention.

Research findings suggest that azole antifungals exert their fungicidal activity in *Aspergillus* by inducing the creation of cell wall carbohydrate patches [26]. These patches, composed of both toxic and nontoxic sterol intermediates, trigger the demise of the fungus [26]. Typically, exposure to effective azole concentrations triggers the generation of carbohydrates patches [26]. Intriguingly, our observations indicate that patch formation can also be prompted by challenging isolates with sub-inhibitory levels of both azoles and LPV. This potential to enhance azole activity, even at sub-inhibitory concentrations, highlights the promising role of LPV as a co-drug. The results demonstrate the efficacy of LPV and azoles in combination and provide valuable insights into potential therapeutic strategies against *Aspergillus* infections.

The investigation of additional *Aspergillus* species, including *A. brasiliensis*, *A. flavus*, and *A. niger*, using the checkerboard assay provided valuable insights into the broader applicability of LPV's synergistic effect in combination with azoles. The observed synergy, reflected by ΣFICI values ranging from 0.13 to 0.38 for POS/ITC in combination with LPV, further

strengthens the potential of LPV as a co-drug to enhance the antifungal activity of azoles across different *Aspergillus* strains.

In conclusion, our study highlights the ability of LPV to revive the antifungal activity of azoles by inhibiting efflux mechanisms. The combination of sub-inhibitory doses of azoles with LPV leads to the formation of carbohydrate patches, signifying enhanced retention of azoles and the accumulation of toxic intermediates. Furthermore, the usage of LPV is well tolerated in a population highly susceptible to *Aspergillus fumigatus* infections [34, 35]. The potential to apply an FDA approved drug to enhance current azole therapies in a clinical setting is readily apparent. Overall, our findings support the use of LPV as a promising antifungal co-drug. Further investigations are warranted to explore the clinical potential of this combination therapy in treating *Aspergillus* infections and combating the emerging challenge of antifungal resistance.

## Materials and methods

### Fungal strains, media, reagents and chemicals

Fungal strains utilized in this study, Table 1, were obtained from the CDC (Atlanta, GA, USA), the American Type Culture Collection (ATCC), and BEI Resources (Manassas, VA, USA). RPMI media components were purchased from Thermo Fisher Scientific (Waltham, MA), RPMI 1640, Sigma-Aldrich (St. Louis, MO) 3-(N-Morpholino) propane sulfonic acid (MOPS), Becton, Dickinson and Company (Franklin Lakes, NJ), Potato Dextrose (PD) broth and agar. Itraconazole and lopinavir were purchased from Acros Organics (New Jersey, USA). Voriconazole was purchased from TCI Ltd., (Tokyo) and posaconazole was purchased from Biosynth Carbosynth (San Diego, CA USA). Rhodamine 6G was procured from Alfa Aesar (Tewksbury, MA USA). Reagents and dyes for fungal cell imaging are as follows: formaldehyde (Sigma Aldrich, St. Louis MO.), triton X-100 (Acros Organics, New Jersey, USA), and calcofluor-white (Sigma-Aldrich, St. Louis, MO).

### Minimum inhibitory concentration (MIC) and microdilution checkerboard assays

To assess the activity of LPV and azoles (ITC, POS and VRC) against a variety of *Aspergillus* species, we conducted a standard broth microdilution checkerboard assay [36, 37]. The MIC and Checkerboard readings were determined at 48 hours post incubation, wells selected for displayed 100% growth inhibition [23]. The fractional inhibitory concentration indexes ($\Sigma$FICI) were calculated to determine drug interactions between the drugs [38]. Drug interactions were determined to be synergistic (SYN) when $\Sigma$FICI $\leq 0.5$, indifferent (IND) at $>0.5$ to $\leq 4$ and antagonistic (ANT) when values were $> 4$ [39–41]. Each assay was independently performed in duplicate; all values were then compared to the published ranges.

### Growth kinetics of *A. fumigatus* CDC #738

To assess the effect of the combinational therapies on *A. fumigatus* strain CDC #738, a growth kinetic was performed [42, 43]. Fungal spores were diluted in RPMI to $5 \times 10^4$ conidia/mL. The fungal cells were then treated with either POS at 0.125x MIC (0.125 μg/mL), ITC at 0.25x MIC (0.25 μg/mL), LPV (16 μg/mL) or a combination of POS and LPV. To monitor the growth, the optical density ($OD_{530}$) of the samples was measured at 0, 6 24, 36 and 48 hours using a SpectraMax i3x microplate reader at 530 nm [43]. VRC at 0.25x MIC (0.125 μg/mL) in combination with LPV at 16 μg/mL was utilized to challenge isolate #738 spores diluted in RPMI at $5 \times 10^4$ conida/mL and read at 0, 24 and 48hrs using a SpectraMax i3x microplate

reader at 530 nm [43]. This allowed us to track the growth progression over 48 hr under the different treatment conditions. The experiment was independently performed in triplicate.

### Efflux pump inhibition in *A*. *fumigatus* by lopinavir

To investigate the mechanism by which LPV enhances azole activity, we performed a rhodamine 6G (R6G) efflux assay [20, 32, 44, 45]. Briefly, mid-log phase cells cultured at 37°C were harvested, washed thrice with PBS, and starved by incubating them in PBS for 8 hr. The starved cells were then incubated with R6G at a final concentration of 7.5 μM for 30 minutes at 37°C and then washed twice with PBS. Fungal cells were treated with LPV (16 μg/mL) then transferred to an opaque 96-well plate, in 100μL aliquots. Glucose at 100mM was added in 25μL amounts to initiate efflux in fungal cells. Over a period of 30 minutes, the fluorescence intensity of R6G was measured at 430/485 nm using a SpectraMax i3x microplate reader (Molecular Devices, San Jose, CA). The fluorescent values were averaged and compared to the control (untreated) wells using unpaired two-tailed *t*-test. This experiment was performed in triplicate.

### Impact of LPV on formation of cell-wall carbohydrate patches induced by azoles

To assess the morphological changes induced by LPV and azole therapy, fungal cells were visualized using calcofluor-white staining as described before [26]. Briefly, *A. fumigatus* strain (CDC #738) was cultured overnight in RPMI 1640 at 37°C in a shaking incubator. The cells were then exposed to either LPV (16 μg/mL), POS (0.125 μg/mL), ITC (0.25 μg/mL), azole/ LPV combination, or left as a negative control. After incubating for 3 hr at 37°C, 200 uL aliquots of the cells were placed onto a slide and allowed to adhere for 3 hr at 37°C. Subsequently, cells were fixed with a solution of 4% formaldehyde and 0.2% Triton X-100, prepared in 1X PBS and incubated at 25°C for 30 minutes. The cells were then washed twice with PBS. For staining, the cells were exposed to 50-100uL calcofluor-white for 10 minutes at room temperature and then washed again twice with PBS. Finally, the stained samples were visualized with a Zeiss LSM 880 confocal laser scanning microscope, and the images were stored in a CZI format for analysis and documentation.

### Acknowledgments

We thank the CDC and BEI Resources for providing us with the *Aspergillus sp*. isolates used in this study.

### Author Contributions

**Conceptualization:** Nicolas Burns.

**Data curation:** Nicolas Burns.

**Formal analysis:** Nicolas Burns.

**Funding acquisition:** Mohamed N. Seleem.

**Investigation:** Nicolas Burns.

**Methodology:** Nicolas Burns.

**Project administration:** Nicolas Burns.

**Validation:** Nicolas Burns.

**Writing – original draft:** Nicolas Burns.

**Writing – review & editing:** Nicolas Burns, Ehab A. Salama, Mohamed N. Seleem.

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
