## [Decision Letter · Decision Letter 0]

6 Feb 2024

PONE-D-24-02276Synergistic potential of lopinavir and azole combinational therapy against clinically important Aspergillus speciesPLOS ONE

Dear Dr. Burns,

Thank you for submitting your manuscript to PLOS ONE. After careful consideration, we feel that it has merit but does not fully meet PLOS ONE’s publication criteria as it currently stands. Therefore, we invite you to submit a revised version of the manuscript that addresses the points raised during the review process.

We look forward to receiving your revised manuscript.

Kind regards,

Felix Bongomin, MB ChB, MSc, MMed, FECMM

Academic Editor

PLOS ONE

Journal Requirements:

2. "We note that your Data Availability Statement is currently as follows:"All relevant data are within the manuscript."

Reviewers' comments:

Reviewer's Responses to Questions

**Comments to the Author**

1. Is the manuscript technically sound, and do the data support the conclusions?

Reviewer #1: Yes

Reviewer #2: Partly

Reviewer #3: Yes

2. Has the statistical analysis been performed appropriately and rigorously? 

Reviewer #1: Yes

Reviewer #2: Yes

Reviewer #3: No

3. Have the authors made all data underlying the findings in their manuscript fully available?

Reviewer #1: Yes

Reviewer #2: Yes

Reviewer #3: No

4. Is the manuscript presented in an intelligible fashion and written in standard English?

Reviewer #1: Yes

Reviewer #2: Yes

Reviewer #3: Yes

5. Review Comments to the Author

Reviewer #1: Please add additional comments for the authors:

General comment:

The paper by Nicolas Burns et al. is about the potential synergistic effect of the antiviral drug lopinavir (LPV) in combination with the azole (ITC and POS) against clinically important Aspergillus species. It evaluates the impact of this combination therapy on A. fumigatus strains and other clinically important Aspergillus species, as well as the potential mechanisms underlying the observed synergistic effects. Additionally, the study addresses the impact of the LPV/azole combination on the hyphal formation, a critical step for fungal dissemination. However, this manuscript exists several limitations. Firstly, the study acknowledges the limitations of LPV's synergistic activity with certain azole drugs, such as voriconazole, and the potential differential response observed between various azoles. Secondly, the potential toxicity of this combination probably exists. Furthermore, the study focus on in vitro combination treatment, and the amount of the strains are limited considering the variability between strains of the same species, and in vivo treatment is still unknown.

The manuscript requires some improvements; see below for detailed comments.

Abstract: In general, methods should be involved in this part.

Introduction: In this part, the reason for choosing LPV combination for treatment should be an independent part and be highlighted.

Line 68: “cyp51A”, use italic.

Line 71:” Non-Cyp51A ” change into “Non-cyp51A ”

Line 80-81: “we focused on LPV and antiviral agent as an enhancer of azoles against

Aspergillus species”. Here can be changed into “we focused on antiviral agent (LPV) as an enhancer of azoles against Aspergillus species”.

Line 77-94: Separate the introduction of LPV and your study content.

Results: In general, many parts belong to methods and discussion or conclusion, when describing the results, there is no need to repeat the methods. Subtitles should not be end with “.”. Figures when mentioned in context should keep identity. The detailed comments are as follows:

Line 97-102 : Here belongs to “Methods and Materials”.

Line 118-120 : “figure 1”, change into “Figure 1”, better put it in bracket.

Line 121-122: Subtitle remove the “.”.

Line 123-126: Here belongs to “Methods and Materials” as well. In addition, what’s the concentration of conidia?

Line 126: “figure 2”, the same as above

Line 136: “fig. 3”, keep identity in context.

Line 138-139: This is the discussion or conclusion.

Line 141-142: before “When”, add “.” to make the sentence completely.

Line 145-149: “figure”, same as above.

Discussion

This section requires some revision.

Line 155-156: “In recognition of this significant concern, the WHO has designated Aspergillosis as a critical priority pathogen”. Aspergillosis is a kind of disease, not a pathogen.

Line 169-176: Is there any literature that mentions about the link between these mutations and the effects of antiviral drugs in combination with azoles? What’s the difference compared with the previous study?

Line 179-181: Why does it work against Candida for the combination of LPV and VRC?

Line 184-185: “Our initial checkerboard data demonstrated that the co-application of LPV enhances azole activity against susceptible strains, as well as two CDC-resistant strains”, add “.” at the end of this sentence.

Line 189-196: This paragraph belongs to “Result”. Add some analysis and references or remove them.

Line 211-215: Concise this part.

Line 216-217: “While ITC and POS can be enhanced by efflux inhibition VRC demonstrated little increased efficacy.”, add “,” after “inhibition”.

Line 216-217: Why the combination of VRC showed different effect?

Line 222-225: Concise into one sentence.

Line 224: Do you think these toxic patches can cause potential damage to cells or humans when applied to clinical therapy?

Conclusion : Here need improvement and condense. Some expressions are not accurate. The study mainly focused on the treatment effects based on the combination of LPV and azoles in vitro. Therefore, there remains a long process from in vitro to in vivo application. In addition, the potential damage after the combination of these drugs probably cause is still unknown.

Line 240-241: This is the reference, not your conclusion for your study.

Method：In general, which method you have referred to must be described and should contain more detailed information in this part. Please list the standard methods or protocols you have chosen for MICs. Also about data analysis, what method was used to analyse? Which software ?

Line 248-249: What is “CDC” and BEI? “table 1” use capital letter for first letter. And put it in bracket.

Line 256: Keep the tense identity.

Line 249-258: Make the format of reagents identity. For example, list the producer and place in the bracket.

Line 260-261: What’s protocol for MICs ?

Line 266-267: What are the published ranges?

Line 269-270: Why here use reference, do you mean that the isolates have been studied? And why you choose this isolate? Why not others or more?

Line 270: Why use this concentration of conidia and is that the final concentration?

Line 270-272: How can you choose the concentrations of different drugs? How do you know this concentration can work?

Line 272-277: Why are the time points setting of POS/ITC and VRC different?

Line 282-288: Did you have any reference for this method? Which strain was applied for this study?

Line 293-294: Why did you choose this concentration of the drugs?

Figure 1: Use different colors for different drugs in the control and tested groups. For example, Red/Purple/Pink contain for ITC/POS/VRC control group, while Blue/Green/Yellow for the LPV combination with ITC/POS/VRC, respectively.

Use a.b.c for sub-figures.

Abbreviations should be interpreted.

Figure 2: The subtitle should be removed in the figures, and it already showed in the legends. The legend should be clear and concise. Abbreviations should be interpreted.

Figure 4: How did you know the patches that were toxic?

Table. 1. About “Source & Characteristic & Resistance Mechanism”, you should add the reference if you did not test in this study, for example, the mutation mentioned in this table.

Reviewer #2: In this manuscript, the authors have studied the synergistic antifungal activity combinational treatment with lopinavir (LPV, an FDA approved antiviral drug) and azoles (antifungal drug) against pathogenic Aspergillus species. They demonstrate that LPV inhibits fungal efflux pump, thereby increasing the efficiency of azoles.

Comments:

1. It is an interesting as well as important study. However, the study reads as if the authors focused on specific activity reported earlier for LPV, i.e., inhibition of efflux pump. It will be interesting also to know whether LPV increases the permeability of fungus facilitating the entry of azoles to execute their function.

2. Methodology of efflux pump inhibition assay is unclear. After transferring fungal cells into opaque plates and adding glucose to initiate efflux pump, in which compartment (inside or outside fungal cells) the florescence was measured and how?

3. Figure 4, particularly 4F is blurred. Better image and with higher magnification should be provided.

4. Data on other Aspergillus species in the form of a figure should be included.

Minor comments (typos):

1. Line 80: Candida should be in italics.

2. Line 120: Figure 1 should be inside parenthesis.

3. Line 142: Put a full-stop after “induced by azoles”.

4. Paragraph 141 to 152: all figures (figure 4a to figure 4f) should be with parenthesis.

5. Line 185: There should be a full-stop after CDC-resistant strains.

6. Line 218: Leave a space between activity and reference numbers.

7. Line 261: Leave a space between assay and reference numbers.

Reviewer #3: Lines 88-94: these sentences are conclusion and should not be mentioned in introduction section.

Method section: where is the statistical analysis section? how did you analyze the results?

Table 1: What are the sources of isolation for CDC 731-740, MYA-3626, MYA-3627 and etc.?

6. PLOS authors have the option to publish the peer review history of their article (what does this mean?). If published, this will include your full peer review and any attached files.

Reviewer #1: **Yes: **Shaoqin Zhou

Reviewer #2: No

Reviewer #3: No

---

## [Author Response · Author response to Decision Letter 0]

18 May 2024

We thank the reviewers for taking their time to reviewer our work and supply us with wonderfully productive and insightful comments. Thank you.

---

## [Decision Letter · Decision Letter 1]

1 Aug 2024

PONE-D-24-02276R1Synergistic potential of lopinavir and azole combinational therapy against clinically important Aspergillus speciesPLOS ONE

Dear Dr. Burns,

Thank you for submitting your manuscript to PLOS ONE. After careful consideration, we feel that it has merit but does not fully meet PLOS ONE’s publication criteria as it currently stands. Therefore, we invite you to submit a revised version of the manuscript that addresses the points raised during the review process.

**Actually, the manuscript was well improved and almost acceptable. Please see the comments below and correct the text. Additional experiment is not needed. **

We look forward to receiving your revised manuscript.

Kind regards,

Daisuke Hagiwara

Academic Editor

PLOS ONE

Journal Requirements:

**Additional Editor Comments:**

Please consider slight modifications below that are raised by editor.

L113-115: please use Figure 1A, 1B, and 1C.

L231: delete a space before”Table 1”

L252, 258: condida->conidia

Reviewers' comments:

Reviewer's Responses to Questions

**Comments to the Author**

1. If the authors have adequately addressed your comments raised in a previous round of review and you feel that this manuscript is now acceptable for publication, you may indicate that here to bypass the “Comments to the Author” section, enter your conflict of interest statement in the “Confidential to Editor” section, and submit your "Accept" recommendation.

Reviewer #1: All comments have been addressed

Reviewer #2: (No Response)

Reviewer #3: All comments have been addressed

2. Is the manuscript technically sound, and do the data support the conclusions?

Reviewer #1: Yes

Reviewer #2: Yes

Reviewer #3: Yes

3. Has the statistical analysis been performed appropriately and rigorously? 

Reviewer #1: Yes

Reviewer #2: Yes

Reviewer #3: Yes

4. Have the authors made all data underlying the findings in their manuscript fully available?

Reviewer #1: Yes

Reviewer #2: Yes

Reviewer #3: No

5. Is the manuscript presented in an intelligible fashion and written in standard English?

Reviewer #1: Yes

Reviewer #2: Yes

Reviewer #3: Yes

6. Review Comments to the Author

**Reviewer #1:** This manuscript has been improved a lot and all comments have been resolved. The manuscript is presented in standard English. And the all data in the manuscript can support their conclusion. In addition, the data analysis is fully rigorous, and all data are available.

**Reviewer #2: **The authors have mostly clarified the queries raised and the modifications suggested. However, looks like no attempts were made to add additional data. At least, the authors can explore whether LPV increases the permeability of the fungus, experimentally, to check whether it is another mechanism facilitating the entry of azoles to execute their function.

**Reviewer #3:** line 87-89: 'Our research revealed that ... observed between POS/ITC and LPV." this sentence should be omitted.

7. PLOS authors have the option to publish the peer review history of their article (what does this mean?). If published, this will include your full peer review and any attached files.

Reviewer #1: **Yes: **Shaoqin Zhou

Reviewer #2: No

Reviewer #3: No

---

## [Author Response · Author response to Decision Letter 1]

18 Sep 2024

We thank the reviewers for their insightful comments and thoughtful edits. We have incorporated them into our work, thank you for your time.

---

## [Editor Report · Decision Letter 2]

24 Sep 2024

PONE-D-24-02276R2Synergistic potential of lopinavir and azole combinational therapy against clinically important Aspergillus speciesPLOS ONE

Dear Dr. Burns,

Thank you for submitting your manuscript to PLOS ONE. After careful consideration, we feel that it has merit but does not fully meet PLOS ONE’s publication criteria as it currently stands. Therefore, we invite you to submit a revised version of the manuscript that addresses the points raised during the review process.

We look forward to receiving your revised manuscript.

Kind regards,

Daisuke Hagiwara

Academic Editor

PLOS ONE

Journal Requirements:

Additional Editor Comments:

L115: Added (Table 2)

L117: Added “figure 1A-C”

L118: Added “figure 1A-B”

L119: Added “figure 1C”

L120: Added “figure 1A-B”

These should be put into parentheses and capitalized at the first character! (This should be applied throughout the text, not used in the required context)

L139-140: This sentence is too strong and overstated. The data shown by the authors only demonstrated that Rhodamine-6G accumulation associated with LPV treatment. The authors can suggest that LPV inhibit the pump function. Accumulation of azole in the cells treated by LPV was not observed in this study. Please rewrite it to appropriate sentence.

L212-213: the sentence sounds a bit wrong. I think "The cells treated LPV showed 70% more accumulation of Rhodamine-6G than the untreated cells."

---

## [Author Response · Author response to Decision Letter 2]

8 Nov 2024

Thank you all for your time and thoroughness with this work.

---

## [Editor Report · Decision Letter 3]

12 Nov 2024

Synergistic potential of lopinavir and azole combinational therapy against clinically important Aspergillus species

PONE-D-24-02276R3

Dear Dr. Burns,

We’re pleased to inform you that your manuscript has been judged scientifically suitable for publication and will be formally accepted for publication once it meets all outstanding technical requirements.

Kind regards,

Daisuke Hagiwara

Academic Editor

PLOS ONE
---

## [Editor Report · Acceptance letter]

18 Nov 2024

PONE-D-24-02276R3 

PLOS ONE

Dear Dr. Burns, 

I'm pleased to inform you that your manuscript has been deemed suitable for publication in PLOS ONE. Congratulations! Your manuscript is now being handed over to our production team.

Kind regards, 

on behalf of

Dr. Daisuke Hagiwara 

Academic Editor

PLOS ONE